# A Non-Rogue Mutant Line Induced by ENU Mutagenesis in Paramutated Rogue Peas (*Pisum sativum* L.) Is Still Sensitive to the Rogue Paramutation

**DOI:** 10.3390/genes12111680

**Published:** 2021-10-23

**Authors:** Ricardo Pereira, José M. Leitão

**Affiliations:** 1Laboratory of Genomics and Genetic Improvement, MED, FCT, Universidade do Algarve, Campus de Gambelas, 8005-139 Faro, Portugal; ricardo.pereira@cortevelada.pt; 2Corte Velada Investimentos, Monte Ruivo, PB 552X, 8600-237 Odiáxere, Portugal

**Keywords:** pea, Pisum, paramutation, rogue phenotype, 1-ethyl-1-nitrosourea, ENU, chemical mutagenesis

## Abstract

The spontaneously emerging rogue phenotype in peas (*Pisum sativum* L.), characterized by narrow and pointed leaf stipula and leaflets, was the first identified case of the epigenetic phenomenon paramutation. The crosses of homozygous or heterozygous (e.g., F1) rogue plants with non-rogue (wild type) plants, produce exclusively rogue plants in the first and all subsequent generations. The fact that the wild phenotype disappears forever, is in clear contradiction with the Mendelian rules of inheritance, a situation that impedes the positional cloning of genes involved in this epigenetic phenomenon. One way of overcoming this obstacle is the identification of plant genotypes harboring naturally occurring or artificially induced neutral alleles, non-sensitive to paramutation. So far, such alleles have never been described for the pea rogue paramutation. Here, we report the induction via 1-ethyl-1-nitrosourea (ENU) mutagenesis of a non-rogue revertant mutant in the rogue cv. Progreta, and the completely unusual fixation of the induced non-rogue phenotype through several generations. The reversion of the methylation status of two previously identified differentially methylated genomic sequences in the induced non-rogue mutant, confirms that the rogue paramutation is accompanied by alterations in DNA methylation. Nevertheless, unexpectedly, the induced non-rogue mutant showed to be still sensitive to paramutation.

## 1. Introduction

The spontaneous emergence of off-type rogue plants in peas (*Pisum sativum* L.), characterized by pointed leaflets and leaf stipula, and the non-Mendelian inheritance of this new phenotype, were described for the first time in the beginning of the last century by Bateson and Pellew [1,2] and soon after confirmed by Brotherton [3,4]. 

The progeny of crosses between rogue and non-rogue (wild) plants is exclusively constituted by “rogues”, a result that would indicate the genetic dominance of the rogue phenotype over the wild phenotype. However, in the F2, and subsequent generations, the wild phenotype is never recovered. Moreover, the cross of F1 (rogue) plants with wild type plants produces also uniquely rogue plants and, again, the wild phenotype is never recovered in the subsequent generations. In fact, the heterozygous plants behave as homozygous, which suggest that in the presence of the rogue allele the non-rogue allele is epigenetically converted into rogue. 

This epigenetic allele alteration was the first reported case of a relatively rare phenomenon, later designated as paramutation [5]

Although the identification of the rogue paramutation has occurred one century ago, no locus involved in this phenomenon has been identified so far. 

Since the initial studies of Bateson and Pellew [1,2] and Brotherton [3,4], very few studies have been carried out aimed at the elucidation of this exceptional phenomenon in peas.

Bunten (1930) [6] confirmed the absence of chromosome differences between rogue and non-rogue plants, while Pyke and Hedley (1984) [7] determined that the reduced size of stipules and leaflets in rogue plants were a consequence of reduced number of cells and not of differences in cell size. The attempts of Mathews (1973) [8] to find correlations between the rogue phenotype and DNA modifications were not successful.

More recently, we obtained the first data regarding the differential expression of some genes in rogues vs. non-rogue plants [9] and found that the rogue phenotype is accompanied by specific alterations of the methylation status of specific genome sequence [10]. Strongly inherited through mitosis, the identified alterations in DNA methylation remained, in multiple cases, conserved through meiosis to the pollen grains, suggesting a transgenerational inheritance of these modifications [10]. 

In maize (*Zea mays* L.), the identification of naturally occurring or induced non-sensitive to paramutation mutant alleles allowed segregating populations to be obtained and genes involved in different paramutations to be identified via map-based cloning. The first recessive mutation affecting the paramutation at the maize *b1* locus (*mop1*—mediator of paramutation 1) was identified as a natural mutation [11]. A second mutation in the same locus (*mop1-2*) was later induced via ethyl methanesulfonate (EMS) mutagenesis [12]. 

The use of EMS mutagenesis has also resulted in the induction of mutations in other genes involved in maize paramutations as the mediator of paramutation2 (*Mop2*) gene [13] and the series of genes required to maintain repression: *rmr1*, *rmr2* [12], *rmr6* [14] and *rmr7* [15]. All these genes were, subsequently, identified via map-based cloning. 

Except for *rmr2*, all the above mentioned genes were found to encode products homolog to proteins involved in siRNA biogenesis and/or RNA directed DNA methylation (RdDM): *mop1*—encodes an RNA-dependent RNA polymerase ortholog of the Arabidopsis RDR2 [16]; *rmr1*—codifies for a protein similar to the Arabidopsis DRD1 and CLSY [17]; *rmr6* is allelic to *mop3* and encodes the first largest subunit of RNA polymerase IV (POL IV) [18,19], *rmr7* was found to be allelic to *Mop2* [13,15] and encodes the shared second largest subunit of both POL IV and POL V [20,21,22].

However, in peas, to the best of our knowledge, since the first reported studies on the rogue paramutation [1] neither the spontaneous reversion to the wild phenotype nor any naturally occurring non-paramutable allele has been identified. 

The identification of a “neutral” allele, insensitive to the paramutagenic effect of the rogue allele, would allow the map-based cloning of the paramutable locus, which would represent a tremendous advance in the elucidation of this amazing phenomenon: the rogue paramutation in peas. 

Here, we describe the induction of a non-rogue mutant of the rogue *cv.* Progreta by ENU (1-ethyl-1-nitrosourea) mutagenesis. 

Termed as SRP1 (suppressor of rogue paramutation 1) the mutated non-rogue phenotype exhibited an extremely unusual inheritance through several generations until its final fixation. Although accompanied by the alteration of the methylation status of two specific genome sequences to a pattern previously identified in non-rogue plants, the non-rogue mutant showed to be still sensitive to the paramutagenic influence of an alien rogue allele.

## 2. Materials and Methods

### 2.1. Plant Material 

Seeds of the rogue cv. Progreta, kindly provided by Mr. Daniel Wherry (UK PULSES Ltd, UK), and seeds of cv. Onward (line JI2722) and derived rogue line (JI2723), kindly provided by Dr. Mike Ambrose (John Innes Institute, UK), were multiplied at the Campus de Gambelas, Universidade do Algarve. The plants were visually inspected for true-to-typeness and the collected seeds were used in further experiments. The cvs. Frilene, Solara and Douce de Provence are, since long years, cultivated in the Campus.

### 2.2. Mutagenic Treatments 

Seeds of cv. Progreta were immersed during 5 min in a disinfection solution containing 10% (v/v) commercial bleach and 0.5% SDS, washed with tap water and germinated in petri dishes over moistened filter paper for 3 days at 24 °C in the dark. 

The mutagenic treatments with the alkylating agent 1-ethyl-1-nitrosourea (ENU) (Sigma Chemical, Co) were performed as previously described for the induction of powdery mildew resistant peas mutants [23]. In the first year, approximately five hundred, 72 h old, seedlings of cv. Progreta were selected, placed inverted into glass beakers and their plumula immersed into a 5 mM ENU solution for 2 h. Seedlings immersed in tap water were used as experimental controls. Treated and control seedlings were planted at the experimental field of the University of Algarve. Due the late spring warm conditions the M1 plants set very few seeds per plant and the M2 seeds were harvested and sown according to a bulk scheme. The M2 plants were visually inspected for induced mutations. 

Similar field experiments were carried out in the two following years, with over seven-hundred seedlings. The better late winter/early spring conditions, allowed a much higher productivity of the M1 plants and a pedigree design was applied, harvesting apart the seeds of each M1 plant and sowing the M2 generation as families of 15–20 seeds.

### 2.3. Plant Cultivation and Plant Crosses

Selected rogue and non-rogue lines, mutant plants and plants originated from controlled crosses, were grown in pots containing 1:1 peat:vermiculite, under glass greenhouse conditions. Crosses between pea lines were carried out under greenhouse conditions as routinely performed for peas [24]. 

### 2.4. Plant Grafting

The shoot apices of grown in pots young seedling of cv. Progreta were cut off and the stem open vertically by the middle. Young shoot apices of recently germinated (4 days old) seedling of cv. Douce de Provence were collected, the stem sharpened on both sides, and inserted vertically into the open stems of cv. Progreta plants. The grafting region was wrapped with parafilm, for protection and tissue regeneration.

### 2.5. DNA Extraction and RAPD and SSR Analyses

Genomic DNA was extracted from full expanded leaves as described in [25]. RAPD analyses were performed as described in [26] using forty Operon Technologies primers (Kits: AA and AM). SSR analyses were performed as described in [27] using the primers for markers AA219, AC58, A9, AD146, AB146, described in [28]. 

### 2.6. Assessment of Differential Methylation

The comparative assessment in cv. Progreta vs. the SRP1 mutant line of the methylation status of 22 sequences previously identified as differentially methylated in cv. Onward vs. its rogue counterpart line JI2723, was performed as previously described [10].

Briefly, equal amounts of leaf genomic DNA of cv. Progreta and SRP1 mutant (M7) plants were pooled apart forming 2 bulks of DNA of two groups of three plants for each genotype. Two micrograms of each DNA bulk were restricted with the enzymes HpaII and MspI, which recognize the same restriction sequence (5’-CCGG-3’) but are differentially sensitive to DNA (cytosine) methylation. Isozyme digestions were performed overnight at 37 °C with 30 U of each enzyme in 20 µL reaction volume, the reactions were stopped by heating the samples at 65 °C for 20 min and the digestion products analyzed by 1.2% agarose gel electrophoresis for 2 h at 8V/cm for evaluation of the expected differential digestion.

One hundred-twenty-five nanograms (125 ng) of each restricted DNA bulk were additionally digested under the same conditions with 1.25 U of EcoRI. An equal volume of a solution containing 5 pMol of EcoRI-adapter, 50 pMol of HpaII/MspI-adapter, 0.5 U of T4 DNA ligase, 2× ligase buffer (Fermentas) was added to each inactivated restriction reaction for ligation of MS-AFLP adapters [10] and the reaction left to proceed overnight at room temperature. 

Amplification primers and PCR conditions for the assessment of the methylation status of the target sequences have been previously described in detail [10].

## 3. Results

### 3.1. Induction and Validation of the Non-Rogue Mutant SRP1

The first chemical mutagenic (ENU) treatments were performed in an already relatively warm late spring, which resulted in low seed set by the M1 plants. For that reason, the M2 seeds were bulk-collected and bulk-sown in the following year. Among several mutant plants, mostly chlorophyll mutants, two out of over 2330 M2 plants exhibited the non-rogue phenotype and were selected and identified as SRP1 and SRP2 (respectively: suppressor of rogue paramutation 1 and 2). 

The unique plant descent of the SRP2 mutant has not set viable seeds and the analysis was further focused on the mutant SRP1 (Figure 1) and its successive generations. 

Much larger mutagenic experiments were repeated in the following 2 years using a pedigree methodology to harvest and sow the M2 seeds. However, no novel non-rogue mutants have been identified This is not surprising due the random nature of the experimental mutagenesis and the need that the induced mutations result in phenotypical alterations. 

To confirm the origin of the putative mutant plants, genomic DNA of the non-rogue mutants, of the wild genotype (cv. Progreta) and other three commercial cultivars grown in the same experimental field: Onward, Solara and Frilene, was extracted and analyzed by RAPD and SSR markers.

Thirty-five out of the forty primers used for RAPD analysis produced amplification products, and among the 217 amplified markers no one was polymorphic between the mutants and the original cv. Progreta, while all the remaining plant genotypes were clearly discriminated (Figure 2; Appendix A). The analysed SSR loci also showed identical homozygous pattern in cv. Progreta and the induced mutants and discriminated between them and the other pea varieties growing in the vicinity (Figure 2). These results allowed to confirm the rogue cv. Progreta as the wild type of the mutant non-rogue plants, and to rule out the hypotheses of seed contamination, uncontrolled crosses with foreign pollen carrying non-paramutable alleles (never identified, so far) or any other handling error.

### 3.2. Inheritance and Fixation of the Induced Non-Rogue Mutation

In contrast to the commonly expected inheritance of a mutation, which if recessive is expected to result in an M3 generation consisting only of plants exhibiting the mutant phenotype or, if dominant is expected to result in an M3 generation exhibiting a 3:1 (mutant to wild) Mendelian segregation, the inheritance of the induced non-rogue phenotype was completely unusual.

In the M3 generation, the identified SRP1 mutant produced 1 plant exhibiting the non-rogue phenotype and 3 plants showing an intermediate phenotype. The non-rogue M3 plant gave rise to a stabilised non-rogue line through all subsequent generations. However, the three M3 intermediate plants exhibited an unexpected segregation producing in the M4 generation, six rogue, three intermediate and three non-rogue plants. Unexpectedly, in the M5 generation all plants emerged as non-rogue, independently of the phenotype of the M4 progenitor (Figure 3). The non-rogue phenotype remained then fixed in all subsequent generations.

Although sharing the non-rogue phenotype, the M5 plants strongly differed by multiple traits as plant size, leaf color intensity, internode length, etc., which, eventually reflect the segregation of other, simultaneously induced, mutations (Figure 4). However, the high variability of multiple phenotypic traits often included non-inherited bizarre features, as unusual and distorted leaf shapes, which suggest strong epigenetic instability (Figure 5).

Despite their strong variability, eventually also resulting from additional mutations, the non-rogue M5 plants, originated from an initial single mutant M2 plant, were hypothetically assumed to share the same mutated allele, which we coined as *srp*1 (*suppressor of rogue paramutation* 1).

### 3.3. Cross of the Non-Rogue Mutant SRP1 Line with Rogue Line JI2723 

Aiming at the establishment of a segregating F2 population for map-based cloning of the *srp*1 gene, direct and reciprocal crosses were performed between multiple SRP1 non-rogue M5 plants and the line JI2723, a rogue paramutant of *cv.* Onward.

As expected, the F1 generation was uniquely constituted by rogue plants, apparently evidencing the genetic dominance of the rogue phenotype. The F2 plants emerged exhibiting a non-rogue phenotype until the third expanded leaf, but immediately evolving to the typical “intermediate” phenotype with the upper leaves exhibiting the rogue phenotype. The F3 plants were all rogues. These results showed that the new non-rogue mutant allele *srp1* is still susceptible to the paramutagenic allele present in the line JI2723. As no segregation of the rogue phenotype was observed in the F2 and following generations, this cross become useless for the positional cloning of the mutated gene.

### 3.4. Differentially Methylated Sequences in the Mutant Line SRP1 vs. cv. Progreta

In a previous publication [10], we reported 22 genomic sequences [29] differentially methylated in rogue vs. non-rogue plants in another (cv. Onward vs. line JI2723) rogue system.

The comparative analysis of the methylation status of these 22 sequences in M6 SRP1 mutant plants vs. the original rogue cv. Progreta, showed that 12 sequences exhibited similar methylation in both genotypes, while 8 sequences did not amplify or exhibited unspecific amplification products. However, two sequences, AAG/AA_325_R and ACT/AG_449_O, were differentially methylated and exhibited in the mutant line the “reverted” pattern, previously identified [10] as specific to the non-rogue epigenotype (Figure 6). 

Although only present in two sequences, the differential methylation in the mutant line confirms that the Pisum rogue paramutation is accompanied by differential methylation of specific genomic sequences.

### 3.5. Is the Rogue Paramutation Maintained by a Mobile Paramutagenic Factor?

The unusual step-by-step fixation of the mutant non-rogue phenotype through three generations until the sudden conversion of all M5 plants to the non-rogue phenotype, raised the working hypothesis that some paramutagenic factor produced in the original rogue genotype (cv. Progreta), but no more produced in the non-rogue mutant, was consecutively diluted through 3 generations until it reached a non-active threshold. 

To identify the presence of such hypothetic factor, assumed to be mobile throughout the plant, apices of non-rogue (cv. Douce de Provence) plants were grafted on the top of young rogue (cv. Progreta) plants. 

Five out of the 10 grafting experiments were successful. However, no modifications were observed in the growing leaves of the grafted genotype, which maintained the characteristic non-rogue phenotype of the original cultivar (Appendix A).

## 4. Discussion

Differently to the observed in maize, naturally occurring non-paramutable, neutral, alleles have never been reported in peas, a circumstance that hampers the use of the positional cloning or genetic association strategies for isolation of gene(s) involved in rogue paramutation. 

To induce non-paramutable alleles of genes required for the establishment and/or maintenance of the paramutation in pea, we have implemented a 1-ethyl-1-nitrosourea (ENU) mutagenesis program using the rogue cv. Progreta, which resulted in the identification of two induced non-rogue mutant M2 plants. 

Although the surviving non-rogue SPR1 mutant plant identified in the M2 generation was expected to be homozygous for the non-rogue mutant allele, 3 out of the 4 M3 plants exhibited an “intermediate” phenotype, a kind of phenotype observed since the first studies of the rogue paramutation [1,2,3,4].

The progenies of the “intermediate” plants (M4 generation) exhibited from the very beginning a clear rogue or non-rogue phenotype or emerged as non-rogue and evolved to rogue after the 5–6th leaf. 

It is noteworthy that, in *Pisum sativum*, the first 5–6 leaf primordia are already present in the shoot apex of dry seeds embryos [30], which suggests that the evolution to the rogue phenotype in “intermediate” plants affect only leaves whose primordia develop after seed germination. 

Induced recessive nuclear mutations in highly homozygous and strongly autogamous diploid plant species as peas, are usually phenotypically identified in the M2 generation, and the successive progenies of the mutated plants exhibit a very similar fixed phenotype through all subsequent generations [23,31]. The rarely induced dominant nuclear mutations exhibit typical Mendelian segregation in the M3 and following generations [31]. In the present case, the slow fixation through 3 additional generations of the mutated non-rogue phenotype and the sudden fixation of the non-rogue phenotype, accompanied by strong variability on other phenotypic traits, in the M5 generation, were, in this regard, extraordinary.

However, is noteworthy that in the maize *b1* paramutation system, among the naturally occurred and the ethylmethanesuphonate (EMS) induced mutations in the *mediator of paramutation1* (*mop1*) locus [12] some *mop1* homozygous plants exhibited somatic instability that resulted in deleterious pleiotropic phenotypes, including delayed flowering, shorter stature, spindly and barren stalks, and feminized tassels [11].

The appearance of intermediate plants that evolve to rogue after the 5–6th leaf, suggests that a factor involved in the establishment of the rogue phenotype needs, after germination, to reach a required level threshold to determine the rogue development of the new leaf primordia. On the other hand, the fixation through additional generations of the mutant non-rogue phenotype suggests that such factor, apparently no more produced due to the homozygous state of the *srp1* mutation, is still present and successively diminishing in concentration through several generations until it reaches a low, non-active, level. 

Among the candidates for such factor, we can hypothesize some sort of small RNA molecule or infectious prion-like protein, which we assumed to be mobile within the plant. Although, the hypothesis of involvement of small RNAs seems the most adequate due the established relationship between paramutation and siRNAs and RdDM [13,16,17] the hypothesis of a prion-like protein is particularly attractive given the spontaneous emergence of the paramutation phenotype, and the remnant effect, in the intermediate plants. However, our preliminary grafting experiments do not corroborate the hypothesis of a paramutagenic factor moving through the plant tissues.

Differences in the penetrance of a paramutation in transgenic polyploid *Arabidopsis thaliana* were recently reported to be caused by differences in the environmental conditions during plant growing. However, this factor cannot explain the different phenotypes during the fixation of the non-rogue phenotype because in our experiments the pea plants were grown side by side, under identical greenhouse conditions [32].

The reversion of the methylation status of two genomic sequences to the expected status in a non-rogue phenotype confirms the previously found differential DNA-methylation in rogue and non-rogue plants [10]. Nevertheless, it remains to be elucidated the importance and role of the differential methylation in this epigenetic phenomenon.

We can conclude that the experimental induction of a non-rogue mutation has demonstrated that the rogue phenotype can be genetically suppressed, and that this phenotype modification is accompanied by the alteration of the methylation status of specific genome sequences, However, the unusual fixation of the induced non-rogue phenotype and the susceptibility of the respective allele (*srp1*) to paramutation in novel crosses, raise new questions regarding this specific paramutation system, which adequate responses can only be found by additional research.

The development of a large-scale comparative gene expression analysis is expected to result in additional insights on the molecular mechanisms that lie beneath this amazing epigenetic phenomenon. However, the identification of naturally occurring or the experimental induction of mutant lines non-sensitive to the rogue paramutation, remains a major specific objective which achievement is expected to lead to the genome mapping and consequent identification of genes involved in establishment and maintenance of the rogue paramutation.

## 5. Conclusions

This section is not mandatory but can be added to the manuscript if the discussion is unusually long or complex.

## Figures and Tables

**Figure 1 genes-12-01680-f001:**
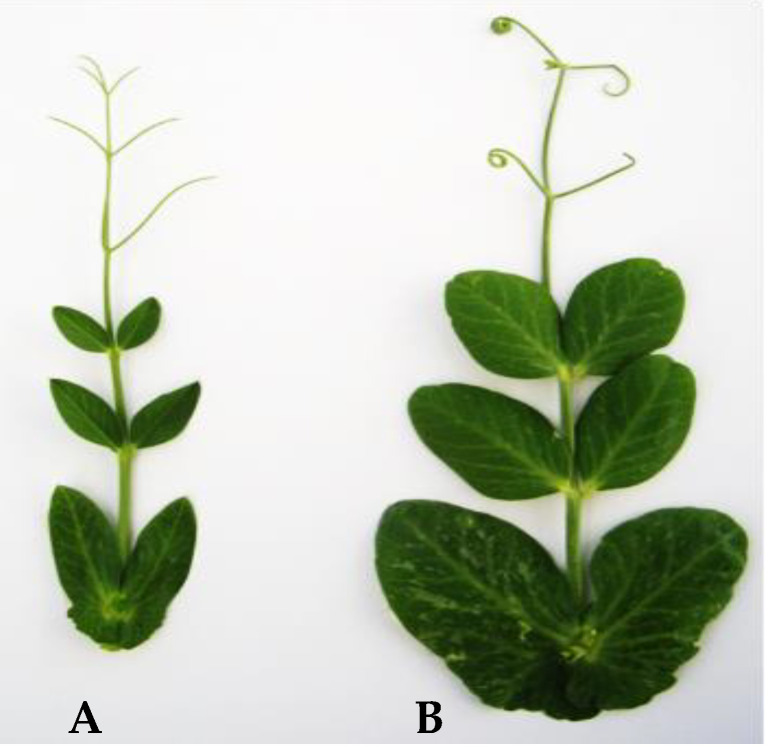
(**A**) Typical leaf of *cv.* Progreta exhibiting the characteristic pointed rogue stipules and leaflets. (**B**) Mutant leaf phenotype of the non-rogue SRP1 mutant.

**Figure 2 genes-12-01680-f002:**
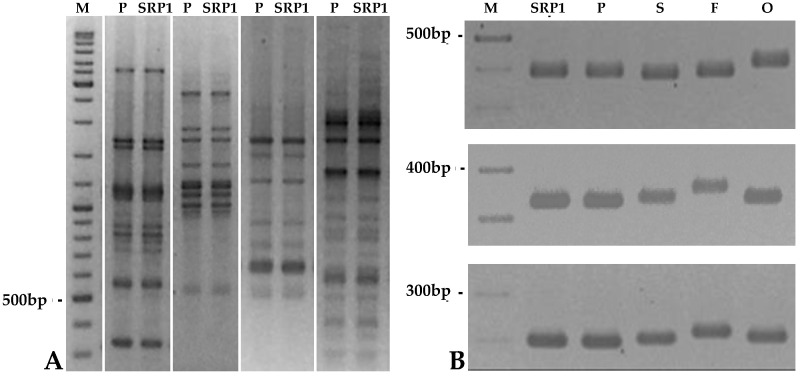
(**A**) RAPD-PCR amplification profiles of genomic DNA of *cv.* Progreta and SRP1 (non-rogue mutant), with primers OPAM07, OPAM09, OPAM10 and OPAM11. (M)—100 bp ladder marker. (**B**) SSR amplification profiles (markers: AD146, AB146, AC58). SRP1 (non-rogue mutant); P—*cv.* Progreta; S—*cv.* Solara; F—*cv.* Frilene; O—*cv.* Onward.

**Figure 3 genes-12-01680-f003:**
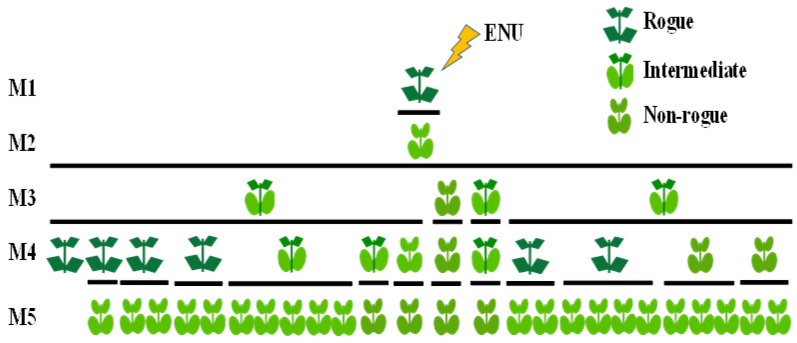
Inheritance of the mutant “non-rogue” phenotype through four generations.

**Figure 4 genes-12-01680-f004:**
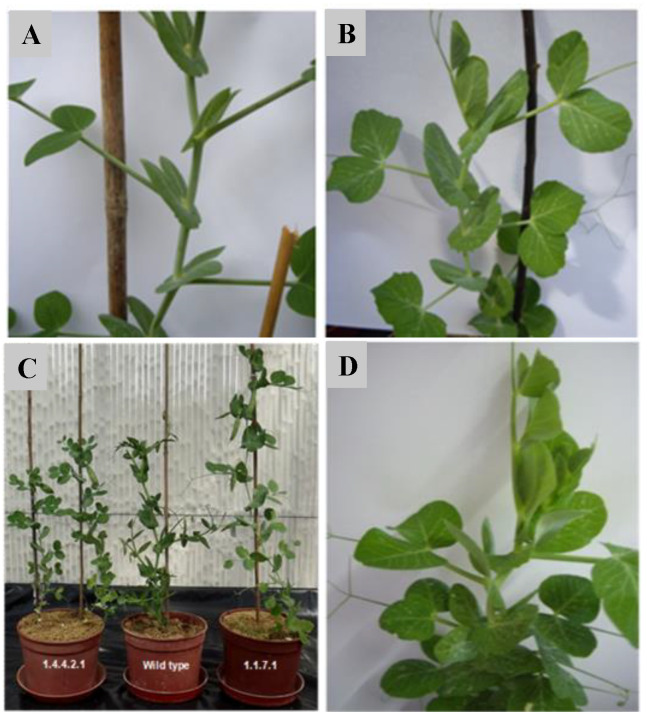
(**A**) M4 plant exhibiting the rogue phenotype. (**B**) M5 plant exhibiting a very clear non-rogue phenotype. (**C**) Left to Right: Two non-rogue M4 plants, one plant of cv. Progreta; an M3 plant of the lineage with fixed non-rogue phenotype. (**D**) M5 non-rogue mutant plant with short internodes.

**Figure 5 genes-12-01680-f005:**
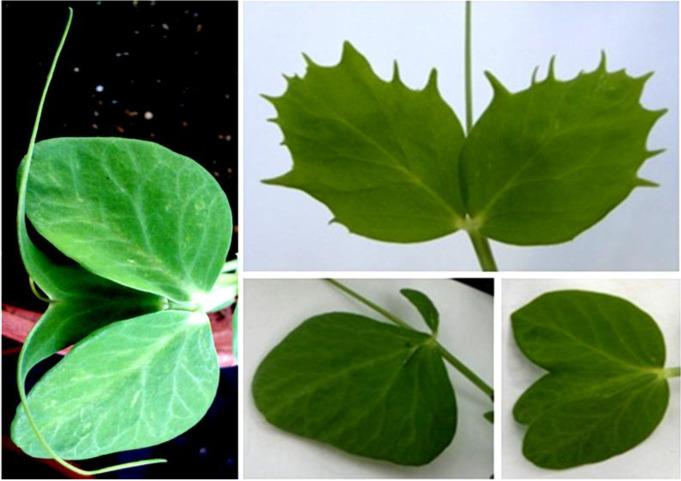
Unusual, non-inherited, tendrils and leaflets exhibited by individual M5 SRP1 plants.

**Figure 6 genes-12-01680-f006:**
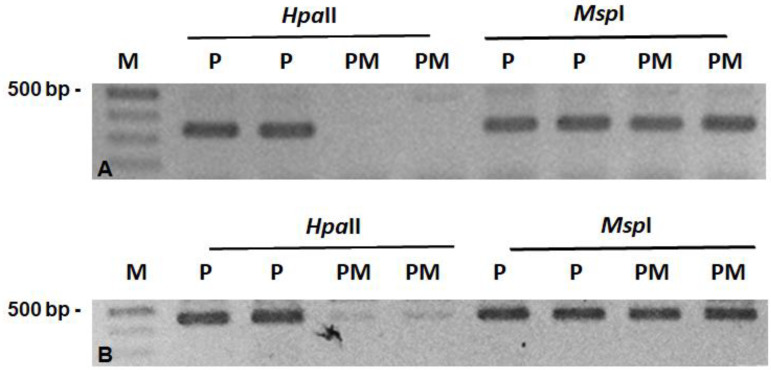
Differential methylation pattern of two specific genomic sequences, AAG/AA_325_R and ACT/AG_449_R in the SRP1 non-rogue mutant line (PM) vs. the rogue *cv.* Progreta (P). In both cases the CpG of the terminal CCGG sequence was methylated by the reversion of the rogue to the non-rogue phenotype. *HpaII* and *MspI*: isoschizomers restriction enzymes differentially sensitive to cytosine methylation.

## Data Availability

Data additional to presented in article and Appendix A are available on request from the corresponding author.

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
