# Peer review of "A Non-Rogue Mutant Line Induced by ENU Mutagenesis in Paramutated Rogue Peas (Pisum sativum L.) Is Still Sensitive to the Rogue Paramutation"

_genes, 2021, doi:10.3390/genes12111680_

Round 1

Reviewer 1 Report

The presented article is devoted to the description of a very interesting phenomenon, i.e., the identification of a mutation that suppresses paramutation in peas. The article is well written and reads with pleasure. However, the reviewer had several questions and comments.

Materials and methods

Why were pea plants inoculated with rhizobia?

Why were macerated nodules of unknown origin used for inoculation, and not a known strain of rhizobia?

What nutrient solution was used to grow the plants?

Results

I think it would be helpful if the authors indicated the frequency of occurrence of rogue mutants.

The authors carried out the mutagenic treatment for 3 years but received rogue mutants only in the first year. Were the authors evaluated the efficacy of mutagenesis in each treatment by analyzing the percentage of chlorophyll mutants so that the 3 mutagenic treatments could be compared?

Is it possible to assume that both obtained mutants are descendants of the same M1 plant, since M2 plants were bulk-collected?

lines 205-206 You wrote “the three M3 intermediate plants exhibited an unexpected segregation producing in the M4 generation, 6 rogue, 4 intermediates and 3 non-rogue plants.” However, on Figure 3 3 intermediate and 4 non-rogue plants are presented.

Authors crossed spr1 mutant with JI2723. I think that it is necessary to cross the spr1 mutant with the original variety and analyze the phenotypes of F1 and F2 generations.

Discussion

I think the inclusion of additional references would be helpful. For example, recently, it has been shown that temperature affects the manifestation of paramutations (Bente H, Foerster AM, Lettner N, Mittelsten Scheid O (2021) Polyploidy-associated paramutation in Arabidopsis is determined by small RNAs, temperature, and allele structure. PLOS Genetics 17(3): e1009444. https://doi.org/10.1371/journal.pgen.1009444).

Author Response

Dear Reviewer

Thank you very much for your comments and recommendations. I hope that we have interpreted and use them correctly.

Reviewer 1

The presented article is devoted to the description of a very interesting phenomenon, i.e., the identification of a mutation that suppresses paramutation in peas. The article is well written and reads with pleasure. However, the reviewer had several questions and comments.

Materials and methods

  1. Why were pea plants inoculated with rhizobia?

2, Why were macerated nodules of unknown origin used for inoculation, and not a known strain of rhizobia?

RESPONSE to questions 1 and 2: The Rhizobia nodules are extracted from roots of strong and healthy pea plants successively grown, by us, in the same greenhouse. The bacteria populations and the plants seem to interact well during several years.  So far, no negative effects have been observed.

We removed this passage from the text, although secondary it can deviate the attention of the readers from more significant facts.

3, What nutrient solution was used to grow the plants?

RESPONSE to question 3. The plants are, routinely, watered twice a week. In one of them the watering is carried out with a 1:8 dilution of the Murashige and Skoog (only mineral) medium.

Results

  1. I think it would be helpful if the authors indicated the frequency of occurrence of rogue mutants.

RESPONSE: Thank you for the suggestion. DONE.

2, The authors carried out the mutagenic treatment for 3 years but received rogue mutants only in the first year. Were the authors evaluated the efficacy of mutagenesis in each treatment by analyzing the percentage of chlorophyll mutants so that the 3 mutagenic treatments could be compared?

RESPONSE: The efficiency of carried out mutation induction experiments is always checked out.  The main parameter is the induction of chlorophyll mutations. This is particularly important in our experiments because the protocol we have developed and regularly use for chemical mutagenesis (Pereira and Leitão doi.org/10.1007/s10681-009-0029-y; Leitão J, 10.1079/9781780640853.0135) results in high survival and low sterility of the M1 plants while increasing the mutagenic efficiency. No significant differences were registered between the three years experiments, despite the different meteorological conditions and breeding approaches.

  1. Is it possible to assume that both obtained mutants are descendants of the same M1 plant, since M2 plants were bulk-collected?

RESPONSE; This is a question to which we cannot give an categorical answer. We do not assume but also, we do not discard, this possibility. This will remain an incognita for us and for the readers.

  1. Lines 205-206 You wrote “the three M3 intermediate plants exhibited an unexpected segregation producing in the M4 generation, 6 rogue, 4 intermediates and 3 non-rogue plants.” However, on Figure 3 3 intermediate and 4 non-rogue plants are presented.

RESPONSE: Thank you very much for calling our attention to this mistake. The figure is right, the description was wrong. The text was corrected accordingly. Please notice that one of the non-rogue plants is descent of a non-rogue M3 plant.

Authors crossed spr1 mutant with JI2723. I think that it is necessary to cross the spr1 mutant with the original variety and analyze the phenotypes of F1 and F2 generations.

RSPONSE: Thank you for your suggestion. This is something that we should do soon. We have not done it before because the main goal was to establish a mapping (highly segregant) population for location of the mutated gene. For that reason, two genetically more distant progenitors were used (the mutant and original rohge line are isolines).

  1. Discussion

I think the inclusion of additional references would be helpful. For example, recently, it has been shown that temperature affects the manifestation of paramutations (Bente H, Foerster AM, Lettner N, Mittelsten Scheid O (2021) Polyploidy-associated paramutation in Arabidopsis is determined by small RNAs, temperature, and allele structure. PLOS Genetics 17(3): e1009444. https://doi.org/10.1371/journal.pgen.1009444).

RSPONSE: We have included a reference to the suggested article in our discussion, However, The penetrance of the rogue phenotypes in our crosses is 100%.  The intermediate phenotype is not inherited. This and other references will be also very useful for the discussion of the gene expression studies that we are presently performing.

Reviewer 2 Report

Dear Author, your manuscript covers a subject that has not been fully explored in pea and should thus be of great interest of Genes readers.

The experiment seems appropriate, and results are properly reported and discussed.

Anyway, English use is not always correct, and some grammatical errors and typoes are spread throughout the manuscript. I have offered some examples.

Sometimes errors in English grammar make the meaning unclear, thus the paper should be edited by a native English speaker to improve English as it would be expected from a paper published in an international journal as Genes.

Lines 169-172 these should be moved to methods and an exaplanation of the cultivar choice should be reported.

References are not edited as requested by the Journal Author 1, A.B.; Author 2, C.D. Title of the article. Abbreviated Journal Name YearVolume, page range.

Sincerely,

Only as examples:

ENGLISH

Line 12, 39 49  wrong accordance subjc-verb

Line 48 aimed to

Line 90 passive: has been shown

Lines 298-300 non usual English – not readable sentence

Line 355, 361 which – whose

TYPOS

Line 13, 352, 353

Author Response

Dear Reviewer

Thank you very much for your comments and recommendations. I hope that we have interpreted and used them correctly.

Reviewer 2

Dear Author, your manuscript covers a subject that has not been fully explored in pea and should thus be of great interest of Genes readers.

The experiment seems appropriate, and results are properly reported and discussed.

Anyway, English use is not always correct, and some grammatical errors and types are spread throughout the manuscript. I have offered some examples.

Sometimes errors in English grammar make the meaning unclear, thus the paper should be edited by a native English speaker to improve English as it would be expected from a paper published in an international journal as Genes.

  1. Lines 169-172 these should be moved to methods and an exaplanation of the cultivar choice should be reported.

Response: We suppose that, for unknown reasons. the lines of the text in your, and our, version do not coincide. The low seed setting of the M1 generation (referred in our text in the cited lines) was a result of the first experimental mutagenesis assays, and we do not see how to move this to the Materials and Methods.

Both the cv. Progreta and the line (JI2723) are characterized as “rogue”. The other cultivars were gown in the vicinity of the mutation induction trials and we needed to discard any influence of these plants (e.g. rare cross-pollination) in the paramutation results-

References are not edited as requested by the Journal Author 1, A.B.; Author 2, C.D. Title of the article. Abbreviated Journal Name YearVolume, page range.

Response: CORRECTED (DONE)

 Sincerely,

 Only as examples:

ENGLISH

Line 12, 39 49  wrong accordance subjc-verb

Line 48 aimed to

Line 90 passive: has been shown

Lines 298-300 non usual English – not readable sentence

Line 355, 361 which – whose

TYPOS

Line 13, 352, 353

RESPONSE: The text was thoroughly revised. We hope to have included the expected corrections.